# No evidence for enhanced disease with human polyclonal SARS-CoV-2 antibody in the ferret model

Douglas S. Reed[1,2]*, Anita K. McElroy[1,3], Dominique J. Barbeau[1,3], Cynthia M. McMillen[1,4], Natasha L. Tilston-Lunel[1,5], Shamkumar Nambulli[1,6], Emily Cottle[1], Theron C. Gilliland[1], Hasala Rannulu[1¤a], Jeneveve Lundy[1], Emily L. Olsen[1¤b], Katherine J. O'Malley[1¤c], Mengying Xia[1¤d], Amy L. Hartman[1,4], Thomas C. Luke[7], Kristi Egland[7], Christoph Bausch[7], Hua Wu[7], Eddie J. Sullivan[7], William B. Klimstra[1,2☉], W. Paul Duprex[1,6☉]

1 Center for Vaccine Research, School of Medicine, University of Pittsburgh, Pittsburgh, Pennsylvania, United States of America, 2 Department of Immunology, School of Medicine, University of Pittsburgh, Pittsburgh, Pennsylvania, United States of America, 3 Division of Pediatric Infectious Disease, Department of Pediatrics, University of Pittsburgh, Pittsburgh, Pennsylvania, United States of America, 4 Department of Infectious Diseases and Microbiology, School of Public Health, University of Pittsburgh, Pittsburgh, Pennsylvania, United States of America, 5 Indiana University–Purdue University Indianapolis, Indianapolis, IN, United States of America, 6 Department of Microbiology and Molecular Genetics, School of Medicine, University of Pittsburgh, Pittsburgh, Pennsylvania, United States of America, 7 SAB Biotherapetuics, Sioux Falls, SD, United States of America

☉ These authors contributed equally to this work.
¤a Current address: School of Medicine, Pennsylvania State University, State College, PA, United States of America
¤b Current address: Department of Microbiology & Immunology, Tulane National Primate Center, Tulane University, New Orleans, LA, United States of America
¤c Current address: WCG Clinical, Inc., Bryn Mawr, PA, United States of America
¤d Current address: Drug Discovery Institute, University of Pittsburgh, Pittsburgh, PA, United States of America
* dsreed@pitt.edu

**Data Availability Statement:** Data will be uploaded to a public repository once manuscript is accepted. For the moment, the data can be access via the

## Abstract

Since SARS-CoV-2 emerged in late 2019, it spread from China to the rest of the world. An initial concern was the potential for vaccine- or antibody-dependent enhancement (ADE) of disease as had been reported with other coronaviruses. To evaluate this, we first developed a ferret model by exposing ferrets to SARS-CoV-2 by either mucosal inoculation (intranasal/oral/ocular) or inhalation using a small particle aerosol. Mucosal inoculation caused a mild fever and weight loss that resolved quickly; inoculation via either route resulted in virus shedding detected in the nares, throat, and rectum for 7–10 days post-infection. To evaluate the potential for ADE, we then inoculated groups of ferrets intravenously with 0.1, 0.5, or 1 mg/kg doses of a human polyclonal anti-SARS-CoV-2 IgG from hyper-immunized transchromosomic bovines (SAB-185). Twelve hours later, ferrets were challenged by mucosal inoculation with SARS-CoV-2. We found no significant differences in fever, weight loss, or viral shedding after infection between the three antibody groups or the controls. Signs of pathology in the lungs were noted in infected ferrets but no differences were found between control and antibody groups. The results of this study indicate that healthy, young adult

following link: https://osf.io/jz3qr/?view_only=
e9060791bb934b54813d2885bbce1831

**Funding:** An NIH award (UC7AI180311) from the
National Institute of Allergy and Infectious Diseases
(NIAID) to WPD supports the Operations of The
University of Pittsburgh Regional Biocontainment
Laboratory (RBL) within the Center for Vaccine
Research (CVR). The funders had no role in study
design, data collection and analysis, decision to
publish, or preparation of the manuscript.

**Competing interests:** HW, TL, CB, KE, and ES are
employees of SAB Biotherapeutics and have
financial interests. This work was supported by a
Department of Defense contract to SAB
Biotherapeutics, Inc. and the University of
Pittsburgh (WK).

ferrets of both sexes are a suitable model of mild COVID-19 and that low doses of specific IgG in SAB-185 are unlikely to enhance the disease caused by SARS-CoV-2.

## Introduction

Coronaviruses (CoV) are positive-stranded RNA viruses of the family *Coronaviridae* in the order *Nidovirales*. Genome size ranges from 26–32 kb. In humans, coronaviruses are mostly associated with the 'common' cold, a mild disease that resolves in 7–10 days. In 2002 a novel coronavirus associated with severe acute respiratory syndrome (SARS-CoV) emerged in Guangdong province, China, with a ∼10% case fatality rate. The virus rapidly spread to other countries [1], but the outbreak was contained, resulting in 8,098 known cases and 774 deaths. A second novel coronavirus associated with severe disease emerged in the Middle East in 2012 (Middle East Respiratory Syndrome; MERS); at 35%, MERS-CoV has the highest case fatality rate of the known zoonotic coronaviruses to date but does not readily spread human-to-human [2]. In late 2019, another novel coronavirus (Severe Acute Respiratory Syndrome-CoronaVirus-2, SARS-CoV-2) associated with severe respiratory disease (COVID-19) emerged in Wuhan, China [3, 4]. This new virus had sequence homology with the original SARS virus and was named SARS-CoV-2. Of the severe CoV, SARS-CoV-2 has the lowest case fatality rate but is the most readily transmissible, which accounts for its rapid spread across the globe. Considerable epidemiologic and virologic data support the notion that asymptomatic or pre-symptomatic individuals infected with SARS-CoV-2 can transmit the virus [5–8]. Risk factors for severe or fatal disease include advanced age and co-morbidities such as heart disease, hypertension, obesity or diabetes [9].

One initial area of concern with SARS-CoV-2 was the potential for pre-existing, virus-specific antibodies, such as those induced by vaccination, to enhance disease upon natural infection by the virus. In studies with feline peritonitis virus, kittens that received a vaccinia-vectored vaccine expressing the spike protein succumbed to challenge earlier than control kittens [10]. Vaccinated kittens had low levels of neutralizing antibody immediately prior to challenge. Passive immunization with antiserum against feline peritonitis virus also caused early death [11, 12]. *In vitro*, virus-specific antiserum increased uptake of the virus by feline macrophages. In another study, mice vaccinated with an inactivated MERS-CoV vaccine had increased eosinophilic lung infiltrates upon challenge [13]. Similarly, mice vaccinated with an alum-adjuvanted SARS-CoV vaccine had enhanced eosinophilia and lung pathology after challenge [14]. Ferrets vaccinated with a modified vaccinia virus Ankara vaccine expressing the spike and nucleocapsid proteins for SARS-CoV had enhanced disease (stronger inflammatory responses & focal necrosis in the liver) when challenged with SARS-CoV [15, 16]. Rhesus macaques vaccinated with an inactivated SARS-CoV vaccine or with vaccinia virus expressing spike protein and challenged with SARS-CoV showed evidence of antibody-enhanced disease [17, 18]. Passive immunization with IgG against SARS-CoV spike abrogated the wound healing process, promoted MCP1 and IL-8 production, and recruited proinflammatory myeloid-lineage cells to the lung. There was cause for concern then that suboptimal antibody response, whether induced by vaccination or passive immunization, could enhance COVID-19 [19].

Ferrets are often used as an animal model for studying respiratory viral infections, particularly influenza, respiratory syncytial virus, and SARS-CoV [20–26]. Relevant to both SARS and SARS-CoV-2, the sequence of the ACE2 receptor in ferrets shares considerable homology with that of humans, suggesting that ferrets could be a potential model for COVID-19 [27]. Other

groups have indeed shown that healthy, adult ferrets can be infected with SARS-CoV-2 and develop mild disease with considerable virus shedding following infection via intranasal and/or intratracheal inoculation [28–30]. There is increasing evidence that airborne transmission of SARS-CoV-2 is important in the spread of COVID-19 but contact transmission is also possible [31–34]. The virus would be more likely to reach the deep lung if inhaled in a droplet (<5 μm mass median aerodynamic diameter) and this might alter disease course and pathology, as we have previously shown for H5N1 [35]. The present study adds to the body of SARS-CoV-2 literature for the ferret model, including different virus isolates, routes of infection, and whether low titers of neutralizing antibody against SARS-CoV-2 spike could enhance disease.

## Materials & methods

### Ethics

The University of Pittsburgh maintains accreditation with the Association for Assessment and Accreditation of Laboratory Animal Care (AAALAC) which require the highest standards for humane conduct of laboratory animal research. The Guide for the Care and Use of Laboratory Animals published by the National Institutes of Health, the Public Health Services Policy on Humane Care and Use of Laboratory Animals, and the Animal Welfare Act guidelines govern all animal work. This work was approved by the University of Pittsburgh Institutional Animal Care and Use Committee (IACUC) prior to initiation of these studies.

### Biological safety

All work with SARS-CoV-2 was conducted in the University of Pittsburgh Center for Vaccine Research (CVR) Regional Biocontainment Lab (RBL) under BSL-3 conditions. Respiratory protection for all personnel when handling infectious samples or working with animals was provided by powered air-purifying respirators (PAPRs; Versaflo TR-300; 3M, St. Paul, MN). Liquid and surface disinfection was performed using Peroxigard disinfectant (1:16 dilution), while solid wastes, caging, and animal wastes were steam sterilized in an autoclave.

### Ferrets

Eleven young adult (4–7 months of age) ferrets of both sexes ranging in weight from 0.8–1.6 kg were purchased from Triple F Farms for use in these studies. In the initial study, all ferrets were male; in the second study half of the ferrets were male, half were female. Ferrets were neutered or spayed and descented before shipment to the University of Pittsburgh.

### Virology

The SARS-CoV-2 isolates used were passage 6 (p6) of the CDC/2019-nCoV/USA_WA1/2020 (WA1) isolate (a gift from Dr. Natalie Thornburg at the Centers for Disease Control in Atlanta) and passage 3 (p3) of the SARS-CoV-2/Münchin-1.1/2020/929 (Munich) isolate (a gift from Drs Christen Drosten & Jan Felix Drexler, Charité –Universitätsmedizin Berlin) described previously [36]. Virus was titrated by conventional plaque assay on Vero E6 cells (ATCC CRL1586) and titers are expressed as plaque forming units (pfu) and infection was visualized in cells by indirect immunofluorescence, as described [36]. Dilutions of virus for plaque assays or inoculations was made in Opti-MEM (Gibco).

### Mucosal inoculation

Ferrets were anesthetized with Isoflurane; once anesthesia was achieved they were inoculated with 1 ml of stock virus given orally, 0.5 ml of stock virus inoculated into each nare, and 0.05

ml of virus inoculated into each eye. Ferrets were returned to their cage and observed until they had fully recovered from anesthesia. For WA1, the challenge dose was $4x10^6$ pfu. For the Munich virus, challenge dose was $8x10^5$ pfu.

## Aerosol exposure

Aerosol exposures were performed using the Aero3G aerosol management platform (Biaera Technologies, Hagerstown, MD). Ferrets were loaded individually into metal exposure cages and transported to a class III biological safety cabinet using a mobile transfer cart. In the class III cabinet, ferrets were placed inside a whole-body exposure chamber and exposed four at a time. Aerosol exposures were 10 minutes in duration. Aerosols were generated using an Aerogen Solo vibrating mesh nebulizer (Aerogen, Chicago, IL) as previously described [37], with a total airflow of 22.0 liters per minute into the chamber. Total exhaust, including sampling, was set to be equal to intake air at the rate of one air change inside the exposure chamber every two minutes. To determine inhaled dose, aerosol sampling was performed during each exposure with an all-glass impinger (AGI) operating at 6 lpm, -6 to -15 psi (AGI; Ace Glass, Vineland, NJ) and containing 10 ml of Opti-MEM and 40 µl of 1:1000 antifoam A (Sigma-Aldrich). Particle size was measured once during each exposure at 5 minutes using an Aerodynamic Particle Sizer with a diluter at 1:100 (TSI, Shoreview, MN). Mass median aerodynamic diameter was 1.72 µm with a geometric square deviation of 1.72. After a 5 minute air wash with clean air, the animal was removed from the cabinet and transported back to its cage. Nebulizer and AGI samples were assessed by plaque assay to determine virus concentration and inhaled dose was calculated as the product of aerosol concentration of the virus and the accumulated volume of inhaled air, determined as the product of the duration of exposure and the animal's minute volume [38]. The inhaled dose of WA1 was $2.3x10^4$ pfu.

## Telemetry

Each ferret was implanted with a DSI PhysioTel Digital radiotelemetry transmitter (DSI Model No. M00) capable of continuously recording body temperature. Telemetry implants were implanted abdominally and the surgical site closed using skin sutures. Ferrets were allowed to heal for 2–4 weeks before challenge. During acquisition, data was transmitted from the implant to a TRX-1 receiver mounted in the room connected via a Communications Link Controller (CLC) to a computer running Ponemah v6.5 (DSI) software. Preexposure data collection began at least seven days in advance of infection. Data collected from Ponemah was exported as 15-minute averages into Excel files which were subsequently analyzed in MatLab 2019a as previously described [39, 40]. Using pre-exposure baseline data, an auto-regressive integrated moving average (ARIMA) model was used to forecast body temperature assuming diurnal variation across a 24-hour period. The code is available at https://github.com/ReedLabatPitt/Reed-Lab-Code-Library. Residual temperatures were calculated as actual minus predicted temperatures. Upper and lower limits to determine significant changes were calculated as the product of 3 times the square root of the residual sum of squares from the baseline data. Fever was defined as any residual temperature (actual temperature minus predicted temperature) which was above the upper limit. ΔTmax was the maximum deviation in residual temperature after infection. Duration was determined as the number of significant temperature elevations divided by 4 to convert to hours. Fever-hours was the sum of significant temperature elevations divided by 4 to convert to hours. Average elevation was calculated as the product of dividing fever-hours by duration.

## Clinical scoring

Clinical signs were recorded twice per day and each animal was given an objective score for weight, temperature, appearance & behavior, and respiratory signs (S1 Table).

## Plethysmography

Respiratory function was assessed in ferrets using a whole-body plethysmography chamber and pneumotach connected to a digital preamplifier run by Finepointe v2.8 software (DSI). For purposes of this study, we used a study protocol for chronic obstructive pulmonary disease (COPD) that is defined in Finepointe software. The chamber and pneumotach were calibrated each day before use. The chamber lid was removed, the ferret was placed in the chamber after which the chamber was sealed. The system was run for 3–5 minutes to acclimate the ferret to the chamber and ensure good data is being collected prior to initiation of data collection which was for 3–5 minutes. Data was analyzed within Finepointe or in Excel & GraphPad.

## Quantitative RT-PCR

RNA was extracted from tissues using Trizol reagent and qRT-PCR assays on extracted RNA were performed exactly as previously described [36].

## SAB-185 human IgG

Production and purification of the SAB-185 human IgG preparation has been described elsewhere [41–45]. Briefly, transchromosomic bovines were hyperimmunized two times with a DNA plasmid expressing the WA-1 spike (S) protein followed by at least three times with purified recombinant S protein of the same amino acid sequence. Bovines were bled and human IgG purified for use in ferret studies.

## Alveolar space analysis

Formaldehyde fixed tissues were paraffin embedded by the University of Pittsburgh McGowan Institute or the Biospecimen Core. Tissue sections were cut to 5 μm and mounted on positively charged glass slides. Standard regressive hematoxylin (Cat # MHS16-500ml) and eosin (Cat # E511-100) staining was used to stain for histopathology. Coverslips were placed on the slides promptly using a 1:1 xylene:toluene mixture as a mounting medium. Using an Olympus Provis microscope, images of stained lungs were captured at 10x for analysis. At least 5 images were randomly selected and captured of the lungs from each ferret. The individual who captured the images and performed the space analysis was blinded to which animals were assigned to which group. Transmitted light was set at 9.0 and exposure time was kept consistent for all images. The microscope was white balanced, and a blank image was obtained to remove shading. NIS-ElementsAR v5.30.01 was used to remove background shading and scale images to 1.35 μm per pixel 47. Alveolar space analysis was performed on H&E samples post imaging in FIJI v1.53c. Images were converted to 8-bit and Huang thresholding with dark background was used to distinguish between alveolar spaces and tissues. Regions of interest were selected and alveolar space was assessed using the 'Analyze Particles' function of the FIJI software with size set to 50 μm2 to infinity to exclude nuclei. Summaries and results for each image were saved as Microsoft Excel files.

# Results

## Comparison of aerosol and mucosal infection with SARS-CoV-2 in ferrets

Prior to evaluating SAB185 for ADE potential, we first developed a ferret model of COVID-19 by infecting ferrets with the WA1/2020 isolate. Because transmission of SARS-CoV-2 is primarily airborne, we compared viral inoculation by the traditional 'mucosal' inoculation (intranasal/oral/ocular) and inhalation of virus via small particle aerosol. Two ferrets were exposed to media alone by either mucosal inoculation or aerosol to serve as mock, uninfected controls (C1 & C2). As expected, there was no fever or substantial (>5%) weight loss in the controls although there is weight loss over time in the controls (Fig 1). Ferrets are prone to lose weight when stressed, and it is possible that the daily monitoring and observations may have been responsible for the weight loss observed. The two mucosally-inoculated ferrets (designated M1 & M2) received a dose of $4\text{x}10^6$ pfu; M1 had a spike in fever within the first 36 hours of infection which returned to normal by 48 hours; there were some small spikes in fever seen at 5–7 dpi and again around 12 dpi. Both mucosally-inoculated ferrets also lost weight fairly rapidly post-infection, reaching 5% around days 2–3 before beginning to rebound. Four ferrets (designated A1-A4) were exposed to WA1/2020 by aerosol with an inhaled dose of $2\text{x}10^4$ pfu. Higher doses by aerosol would only have been possible by generating a very large stock of virus and concentrating it via sucrose-gradient purification. Because of concerns regarding adaptation of the virus to cell culture and the significant amount of time and cost required, concentrating the virus was not pursued. No significant fever was seen in the 4 ferrets exposed to WA1/2020 by aerosol and only 1 of the 4 lost >5% of its body weight in the post-infection period. There was also no significant change in respiratory function or clinical scores recorded during the post-infection period compared to the uninfected controls (S1 and S2 Figs).

Viral shedding from mucosal sites was also assessed in these ferrets (Fig 2). In the mucosally-inoculated ferrets, vRNA (viral) was found on oral swabs taken on days 2 and 4 from both ferrets. Nasal swabs were positive on days 2 and 7 from both mucosal-inoculated ferrets and one of the two on day 10. For both mucosal-inoculated ferrets, vRNA was isolated from rectal swabs only at 2 dpi. Infectious virus was isolated from only M1 at 2 dpi in the oral and nasal cavities and day 4 orally from both M1 and M2 (S2 Table). Despite the lower dose achieved by aerosol, viral shedding from the oral and nasal cavities was very similar for aerosol-infected ferrets compared to mucosal-inoculated ferrets in terms of genomes/ml and duration. Oral swabs from three of the four aerosol-infected ferrets were positive for vRNA out to day 7; two of the four were also positive for vRNA in the nasal cavity at day 7. Two of the four aerosol-infected ferrets were positive for vRNA from rectal swabs on day 4 and one was positive on day 7. Infectious virus was found in the oral and nasal cavities from all four aerosol-infected ferrets on day 2 but only two were positive on day 4 and one on day 7 (S2 Table). None of the mucosally-inoculated or aerosol-infected ferrets were positive for infectious virus from the rectal swabs at any timepoint.

Ferrets were euthanized after day 14 post-infection with WA1/2020 and samples collected to evaluate viral load in tissues. No infectious virus was found in any tissues. For the mucosally-inoculated ferrets, vRNA was found sporadically throughout the respiratory tract, intestines, lymph nodes and also the CNS, heart, kidney and bladder (Fig 3). Lower levels of vRNA were found in the aerosol-infected ferrets, and was mostly confined to the respiratory tract.

## No evidence for antibody-dependent enhancement of disease with SAB185

SAB185 is human IgG specific for SARS-CoV-2 that is purified from the plasma of vaccinated transchromosomic cows expressing human immunoglobulin genes. As part of the pre-clinical

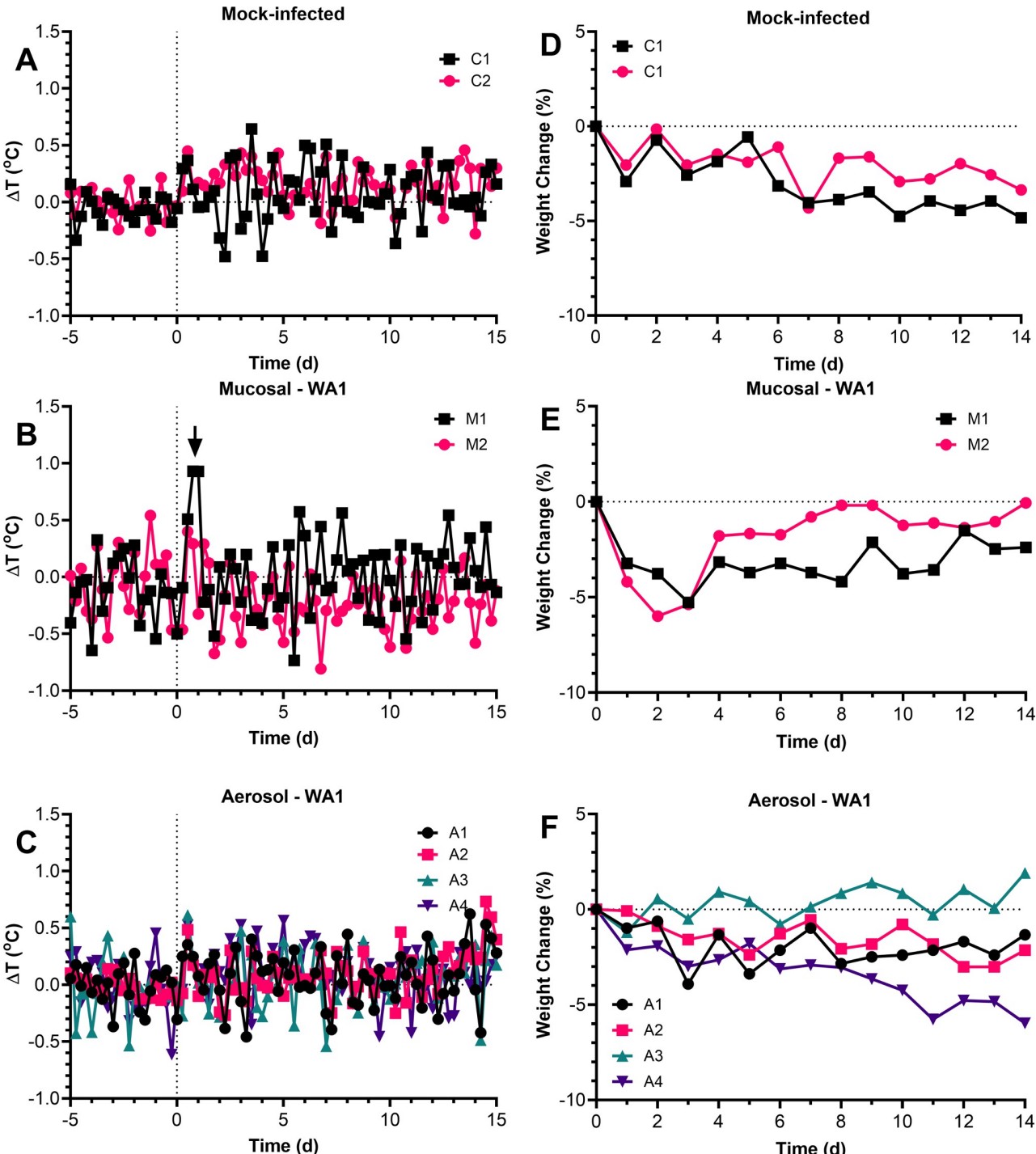

**Fig 1. Mild fever and weight loss in ferrets infected with WA1/2020 either mucosally or by aerosol.** Graphs show residual 6-hour median body temperatures (difference between actual and predicted temperatures) (A-C) for individual ferrets in each group as determined from ARIMA modeling. Graphs in D-F show percent change in body weight (D-F) in control mock-infected ferrets (D), mucosally-inoculated ferrets (E) and aerosol-infected ferrets (F).

evaluation of SAB185, we examined whether low doses of SAB185 would induce enhanced disease in ferrets when challenged with SARS-CoV-2. Based on the model development data, we opted to go with the mucosal inoculation, but with a European isolate that had the D614G

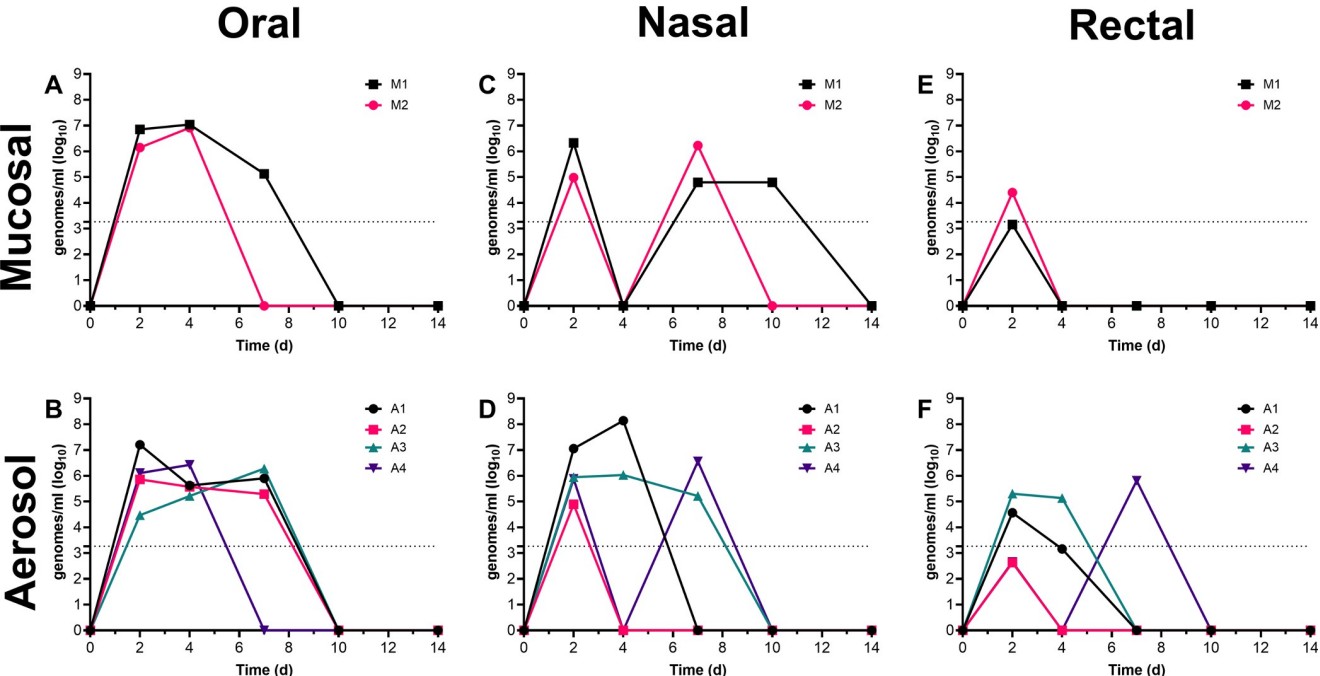

**Fig 2. Viral shedding from ferrets after mucosal or aerosol infection with WA1/2020.** Oral, nasal, and rectal areas of ferrets were swabbed on days 2, 4, 7, 10 and 14 after infection with WA1/2020. Data shown is in genomes/ml determined from qRT-PCR for ferrets infected with WA1/2020 via mucosal (top) and aerosol (bottom) delivery of virus from A, B) oral swabs, C, D) nasal swabs, and E, F) rectal swabs. The limit of detection for viral RNA by PCR was 1,856 genomes/ml, denoted as the dashed lines on the graphs.

mutation (Munich-1.1) instead of WA1/2020. Twelve hours before challenge, ferrets were inoculated IV with SAB185 at doses of 0.1, 0.5, or 1.0 mg/kg. A fourth group was given an irrelevant control IgG against dengue virus. The following morning, all of the ferrets were challenged with $8 \times 10^5$ pfu of Munich-1.1 by mucosal inoculation (intranasal/oral/ocular).

Similar to the WA1/2020 isolate, control ferrets inoculated mucosally with Munich-1.1 spiked a fever 1–2 dpi which subsided within a day although occasional spikes in temperature were seen out to 14 dpi when the study was ended (Fig 4). Similar spikes were seen in all three SAB185 groups but in several of the ferrets from these groups there was a more substantial spike in temperature between 6–10 dpi than was seen in the controls. However, quantitative analysis of the fever response failed to identify a significant difference in the fever response between groups in terms of maximum temperature seen, duration or severity (S3 Table). Overall, average daily weight loss in all four groups was minimal (less than 5% from baseline) although there was one ferret in the control group that lost nearly 10% of its body weight in the first 3 days after infection but then quickly recovered (Fig 5). One ferret in the 1.0 mg/kg group had a more gradual weight loss over the 14 days post-infection but by study endpoint had only lost 5% of its baseline weight. Similar to what was seen in the pilot study, there were no significant changes in respiratory rate for any of the ferrets nor any significant changes in clinical score between control ferrets and SAB-inoculated ferrets (S3 and S4 Figs).

## No change in viral shedding with SAB185 treatment

We also examined whether low dose SAB-185 treatment altered the kinetics of viral shedding in the ferrets. Fig 6 shows the levels of vRNA recovered from oral swabs (A), nasal washes (B) and rectal swabs (C) out to 14 days after infection. At all three sites, vRNA titers were higher

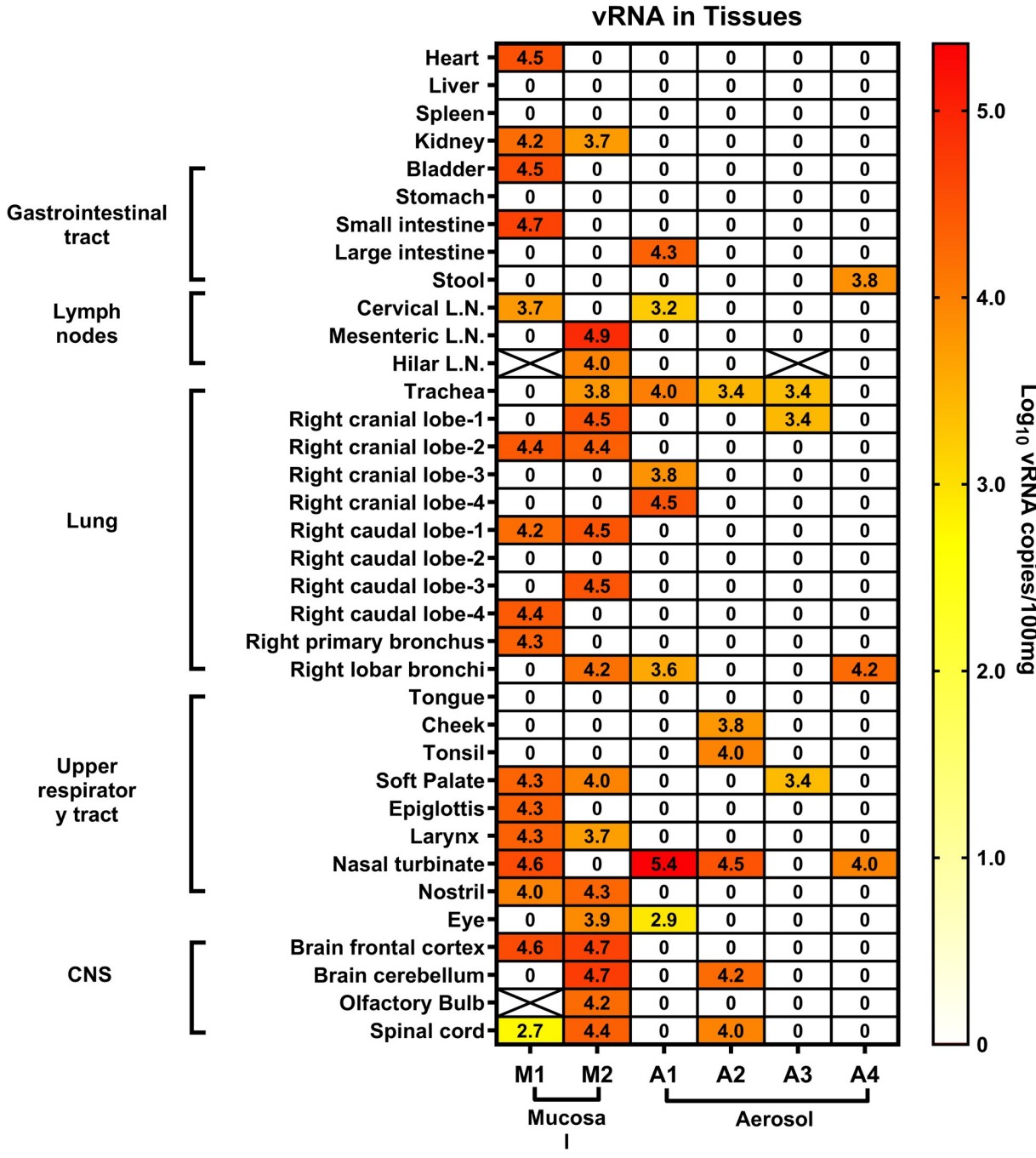

**vRNA in Tissues**

| | M1 | M2 | A1 | A2 | A3 | A4 |
|---|---|---|---|---|---|---|
| Heart | 4.5 | 0 | 0 | 0 | 0 | 0 |
| Liver | 0 | 0 | 0 | 0 | 0 | 0 |
| Spleen | 0 | 0 | 0 | 0 | 0 | 0 |
| Kidney | 4.2 | 3.7 | 0 | 0 | 0 | 0 |
| Bladder | 4.5 | 0 | 0 | 0 | 0 | 0 |
| Stomach | 0 | 0 | 0 | 0 | 0 | 0 |
| Small intestine | 4.7 | 0 | 0 | 0 | 0 | 0 |
| Large intestine | 0 | 0 | 4.3 | 0 | 0 | 0 |
| Stool | 0 | 0 | 0 | 0 | 0 | 3.8 |
| Cervical L.N. | 3.7 | 0 | 3.2 | 0 | 0 | 0 |
| Mesenteric L.N. | 0 | 4.9 | 0 | 0 | 0 | 0 |
| Hilar L.N. | ✕ | 4.0 | 0 | 0 | ✕ | 0 |
| Trachea | 0 | 3.8 | 4.0 | 3.4 | 3.4 | 0 |
| Right cranial lobe-1 | 0 | 4.5 | 0 | 0 | 3.4 | 0 |
| Right cranial lobe-2 | 4.4 | 4.4 | 0 | 0 | 0 | 0 |
| Right cranial lobe-3 | 0 | 0 | 3.8 | 0 | 0 | 0 |
| Right cranial lobe-4 | 0 | 0 | 4.5 | 0 | 0 | 0 |
| Right caudal lobe-1 | 4.2 | 4.5 | 0 | 0 | 0 | 0 |
| Right caudal lobe-2 | 0 | 0 | 0 | 0 | 0 | 0 |
| Right caudal lobe-3 | 0 | 4.5 | 0 | 0 | 0 | 0 |
| Right caudal lobe-4 | 4.4 | 0 | 0 | 0 | 0 | 0 |
| Right primary bronchus | 4.3 | 0 | 0 | 0 | 0 | 0 |
| Right lobar bronchi | 0 | 4.2 | 3.6 | 0 | 0 | 4.2 |
| Tongue | 0 | 0 | 0 | 0 | 0 | 0 |
| Cheek | 0 | 0 | 0 | 3.8 | 0 | 0 |
| Tonsil | 0 | 0 | 0 | 4.0 | 0 | 0 |
| Soft Palate | 4.3 | 4.0 | 0 | 0 | 3.4 | 0 |
| Epiglottis | 4.3 | 0 | 0 | 0 | 0 | 0 |
| Larynx | 4.3 | 3.7 | 0 | 0 | 0 | 0 |
| Nasal turbinate | 4.6 | 0 | 5.4 | 4.5 | 0 | 4.0 |
| Nostril | 4.0 | 4.3 | 0 | 0 | 0 | 0 |
| Eye | 0 | 3.9 | 2.9 | 0 | 0 | 0 |
| Brain frontal cortex | 4.6 | 4.7 | 0 | 0 | 0 | 0 |
| Brain cerebellum | 0 | 4.7 | 0 | 4.2 | 0 | 0 |
| Olfactory Bulb | ✕ | 4.2 | 0 | 0 | 0 | 0 |
| Spinal cord | 2.7 | 4.4 | 0 | 4.0 | 0 | 0 |

Tissue groupings: Gastrointestinal tract (Stomach, Small intestine, Large intestine, Stool); Lymph nodes (Cervical L.N., Mesenteric L.N., Hilar L.N.); Lung (Trachea, Right cranial lobe-1 through 4, Right caudal lobe-1 through 4, Right primary bronchus, Right lobar bronchi); Upper respiratory tract (Tongue, Cheek, Tonsil, Soft Palate, Epiglottis, Larynx, Nasal turbinate, Nostril, Eye); CNS (Brain frontal cortex, Brain cerebellum, Olfactory Bulb, Spinal cord).

Columns M1, M2 = Mucosal; A1, A2, A3, A4 = Aerosol.

Color scale: Log$_{10}$ vRNA copies/100mg (0 to 5.0)

**Fig 3. Viral RNA is predominantly found in the respiratory tract and gastrointestinal system of SARS-CoV-2-infected ferrets.** Results shown are in log$_{10}$ genomes/100 mg of WA1/2020 in the tissues of the two mucosal-infected ferrets and four aerosol-infected ferrets. Tissues were collected at day 14, the study endpoint.

than those seen in the WA1/2020-infected ferrets and persisted longer. No significant differences in vRNA levels were seen between the controls and any of the SAB185 groups, suggesting that low-dose SAB185 neither prevented nor enhanced viral shedding/transmission. Infectious virus was also found in the oral swabs and nasal washes from ferrets in all groups out to day 6 post-infection (S4 Table) although viral RNA was found in all groups out to day 8 for nasal

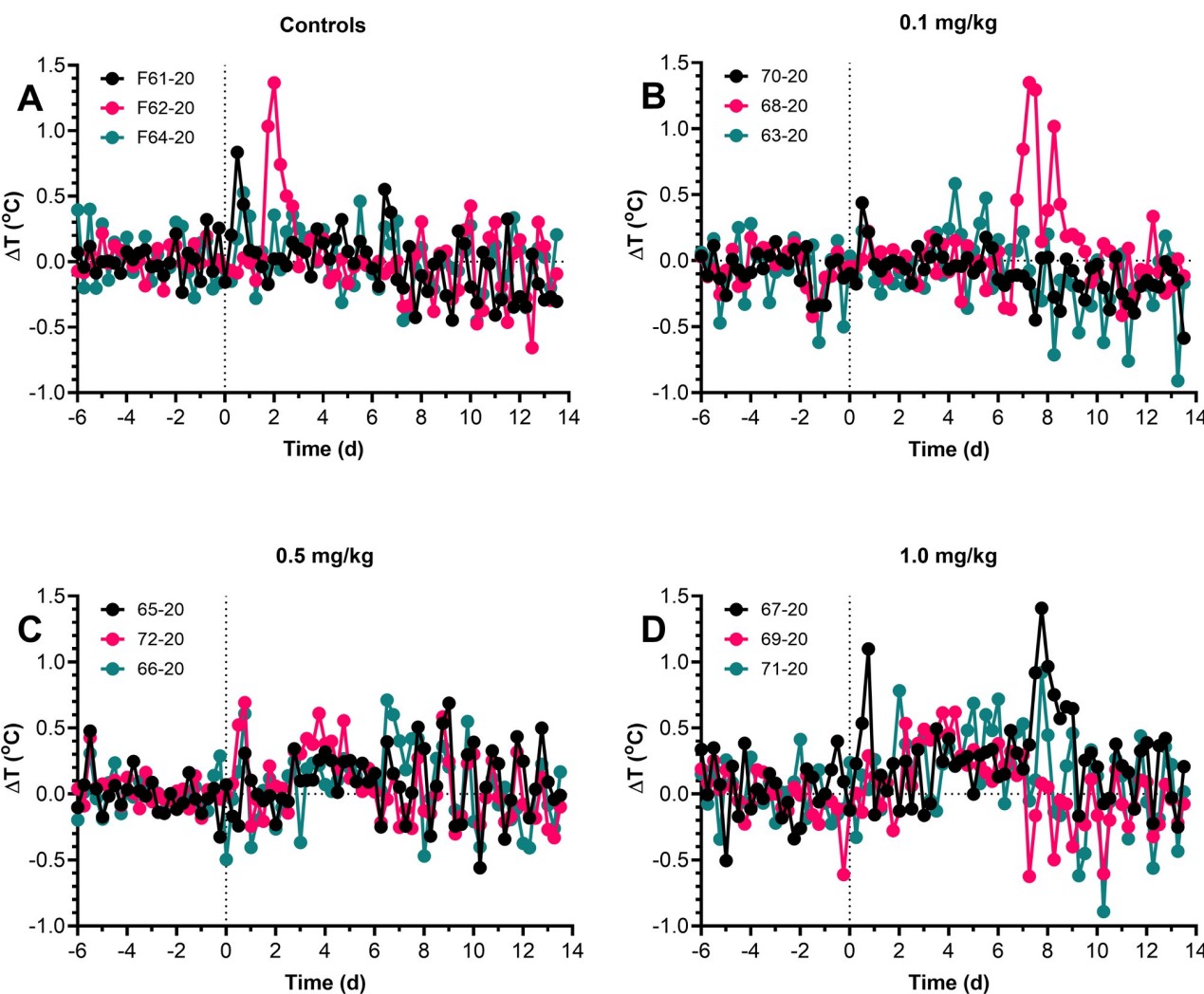

**Fig 4. No difference in fever response to mucosal Munich SARS-CoV-2 challenge in SAB185-inoculated ferrets.** Ferrets were inoculated with SAB185 intravenously 1 day prior to challenge with the Munich virus. Graphs show residual 6-hour median body temperatures (difference between actual and predicted temperatures) (A-D) for individual ferrets in each group as determined from ARIMA modeling after mucosal infection with Munich. The numbers in the symbol legends are the individual ferret identification numbers. A) Control ferrets, B) ferrets inoculated with 0.1 mg/kg SAB-185, C) 0.5 mg/kg SAB-185, D) 1.0 mg./kg SAB-185.

and rectal swabs and out to day 15 for oral swabs from the controls and 0.1 mg/kg groups. Rectal swabs were negative for infectious virus at all timepoints and in all ferrets.

### No evidence for antibody-enhancement of lung pathology

Lungs collected at study endpoint were also evaluated for pathological changes by hematoxylin and eosin staining, with representative images shown in Fig 7. Despite the general lack of overt clinical signs of disease, there was considerable pathology noted in the lungs of all of the infected ferrets. There does not appear to be any enhanced pathology in SAB185-inoculated ferrets at any of the doses tested. Alveolar space analysis of multiple 10x images of lungs from each ferret are shown in Fig 7B. Although the pathology appears worse for the Munich isolate compared to WA1 in the images shown, the alveolar space analysis of multiple images from the lungs of infected ferrets found no significant difference between Munich and the mucosal

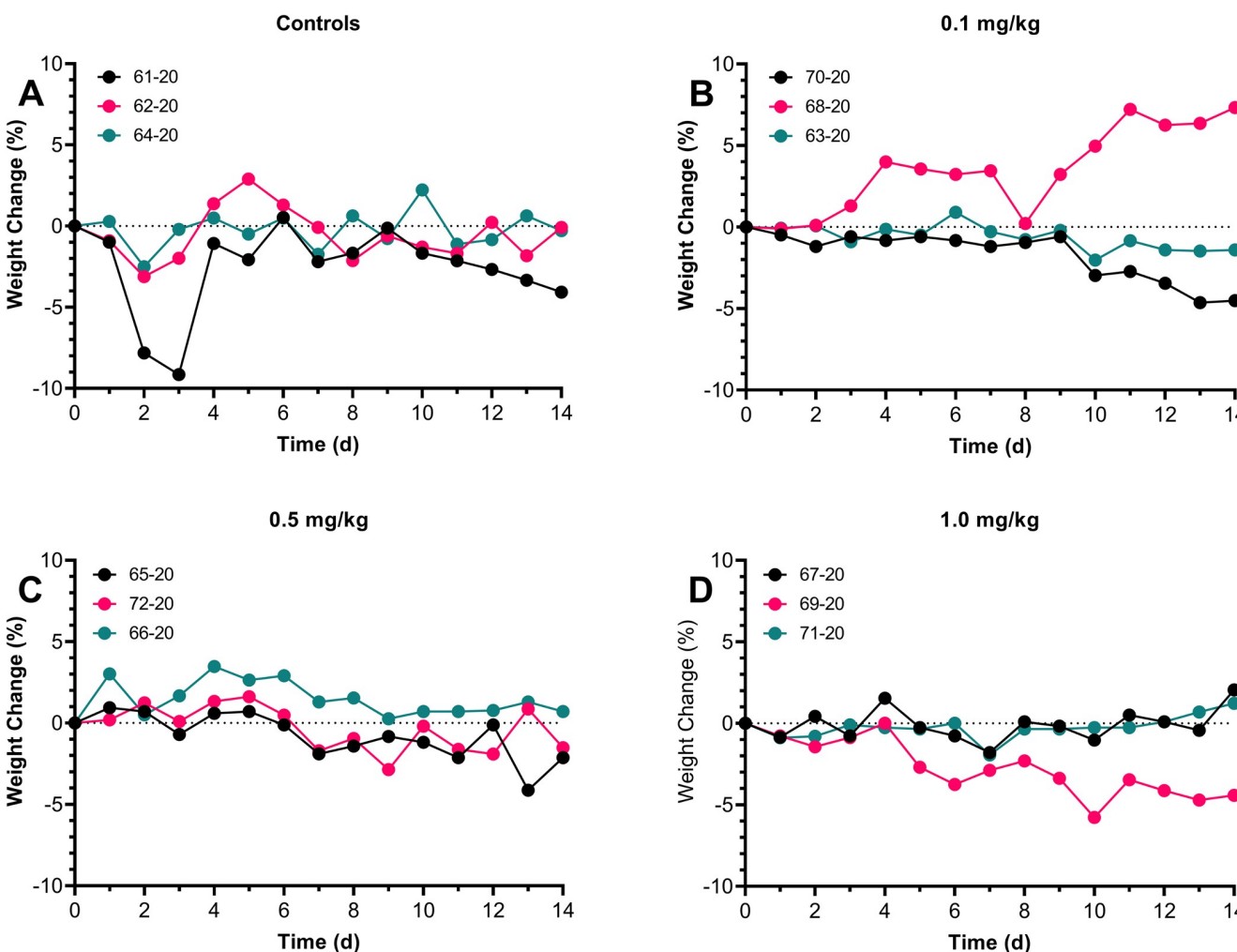

**Fig 5. Minimal weight loss after mucosal infection with Munich SARS-CoV-2 in SAB-185 inoculated ferrets, regardless of dose.** Ferrets were inoculated with SAB185 intravenously 1 day prior to mucosal challenge with the Munich virus. Graphs shown daily percent change in weight from baseline, pre-challenge weights. The numbers in the symbol legends are the individual ferret identification numbers. A) Control ferrets, B) ferrets inoculated with 0.1 mg/kg SAB-185, C) 0.5 mg/kg SAB-185, D) 1.0 mg./kg SAB-185.

WA1 group. There appears to be more alveolar space in the ferrets infected with WA1 by aerosol, but this difference was not statistically significant (likely because of the number of animals used and the range of values seen). The apparent difference between aerosol and mucosal could be a function of the virus dose, since the aerosol exposure dose was 100-fold lower than what was given mucosally. Alveolar space analysis found no differences in ferrets inoculated with SAB-185 and challenged with Munich compared to control (no antibody) Munich-infected ferrets. What is notable is that across all the groups, in many of the ferrets the pathology is not uniform within an individual animal, with some areas showing minimal pathology and others showing considerable pathology. Some regions have 80–90% alveolar space and other regions in the same ferret have less than 20% alveolar space.

## Discussion

We have reported here our efforts to develop the ferret as an animal model for COVID-19 and to use that model to evaluate the potential for antibody-enhanced disease. Ferrets were initially

**Oral**

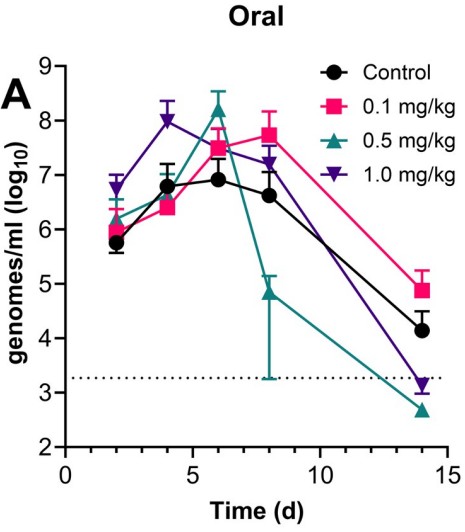

**Nasal**

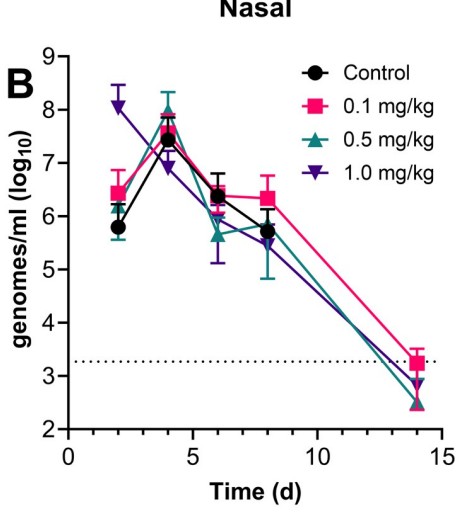

**Rectal**

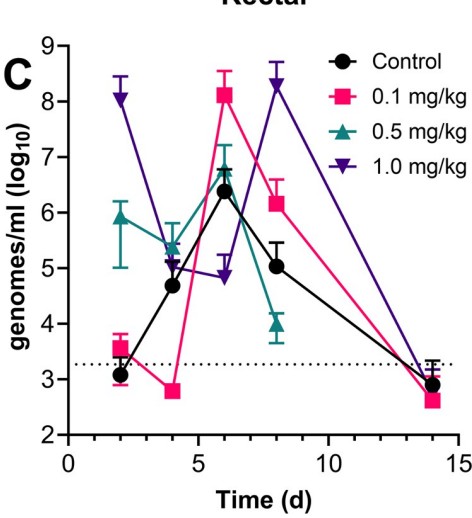

**Fig 6. Viral shedding from mucosal sites is not affected in SAB-185 inoculated ferrets after mucosal Munich challenge.** Ferrets were inoculated with SAB185 intravenously 1 day prior to mucosal challenge with the Munich virus. Oral, nasal, and rectal areas of ferrets were swabbed on days 2, 4, 7, 10 and 14 after challenge. Graphs show average and standard deviation of viral titer as determined by qRT-PCR at mucosal sites over time for each group as determined from A) oral, B) nasal, and C) rectal swabs after challenge. The limit of detection for viral RNA by PCR was 1,856 genomes/ml, denoted as the dashed lines on the graphs.

thought to be a promising model based on prior work done with the original SARS-CoV and similarities in the ACE2 receptor between human and ferret [30]. A number of groups have reported efforts in the last four years to infect ferrets with SARS-CoV-2 [28–30, 46–54]. As would be expected, there are differences in the particular virus isolate used, how the virus stock was generated, the virus dose given, the route of inoculation, or the clinical signs used to determine disease. The reports have two things in common: 1) no severe disease/death in ferrets infected with circulating viruses, and 2) detectable virus replication in the respiratory tract and other mucosal sites. Only one other report has directly compared aerosol and mucosal inoculation of ferrets with SARS-CoV-2 [50]. In that report, SARS-CoV-2 infection was asymptomatic regardless of route. Other groups have also reported asymptomatic infection in ferrets [30, 51, 54] while others reported mild or moderate disease [46, 47, 52]. In our studies we did not observe any overt clinical signs of disease, however we did detect a fever in some of the infected ferrets via the implanted radiotelemetry devices. The fever response was mild and did not persist, although in a small number of ferrets we did see a late febrile response that was stronger and lasted for 2–3 days. None of the other studies directly evaluated the possibility of AED although one study did evaluate whether re-infection was possible and whether vaccination could prevent infection or disease [53]. In that study, they did not report any findings that would suggest AED had occurred.

While conducting these studies, the D614G mutation in the spike protein emerged in Europe and quickly became the dominant circulating virus around the world. We had

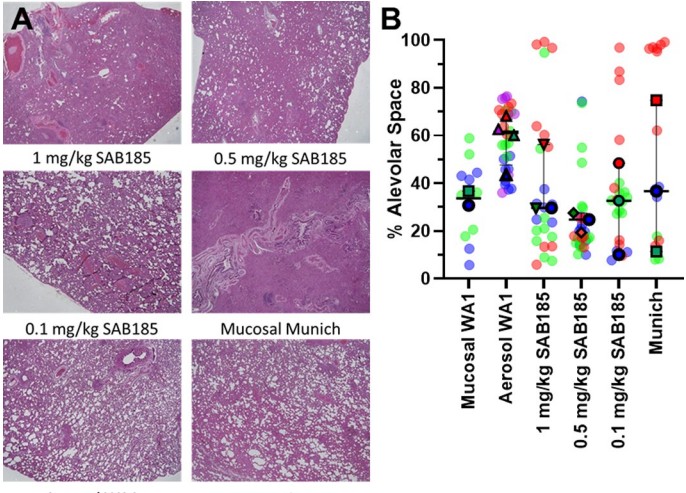

**Fig 7. Low dose SAB185 does not affect lung pathology of ferrets infected with Munich SARS-CoV-2.** Ferrets were inoculated with SAB185 intravenously 1 day prior to mucosal challenge with the Munich virus. A) hematoxylin and eosin staining of representative lung sections from each group, 2x magnification. B) superplot of alveoloar space analysis from lung sections imaged at 10x. The different colors in each group represent individual ferrets. Multiple 10x images of the lung from individual ferrets within a group are shown as the same color without borders; the mean for that ferret is the colored symbol with the black border. The black line and error bars are the median value with the interquartile range for that group.

obtained a SARS-CoV-2 isolate from Munich which had the D614G mutation. While the WA1/2020 and Munich-1.1 isolates were not tested head-to-head in the same study, the data we obtained in ferrets demonstrated more disease (fever, weight loss) and prolonged viral shedding with Munich-1.1 compared to WA1/2020 even though the dose of Munich-1.1 given was lower. At necropsy, we found evidence for the persistence of viral RNA for WA1/2020 (but not infectious virus) out to at least day 14. This is similar to what we saw in nonhuman primates where viral RNA was isolated from African green monkeys more than a month after infection [55]. In the African green monkeys, we also saw occasional fever spikes that coincided with increased viral RNA at mucosal sites 2–3 weeks after resolution of the original infection. Whether these findings in ferrets and African green monkeys is just remnants of viral RNA or intact virus is not clear but this may warrant further investigation considering reports of chronic COVID-19 in patients with initially mild/moderate disease.

We also partnered with SAb Biotherapeutics to evaluate whether SAB185, a polyclonal humanized antibody product for treatment of SARS-CoV-2 infection, would enhance disease if given at a low dose. Prior published data from other coronaviruses in animal models, including ferrets, have shown evidence for antibody-enhanced disease after vaccination or passive immunization [10–17]. Systemic inflammation, hepatitis, and even death have been observed. Since SARS-CoV-2 produces only mild disease in ferrets [29, 30], we reasoned that if low doses of antibody induced enhanced disease, ferrets might be a good model for making that determination. Our results (fever, weight, viral shedding, pathology), however, would suggest that at least with SAB185 in ferrets at the doses administered, there are no concerns about ADE. This data was submitted as part of the IND request that was submitted to, and allowed, by the FDA for the ACTIV2 SARS-CoV-2 clinical trial (ClinicalTrials.gov Identifier: NCT04469179). It should be noted that at least one study has suggested rapid clearance of a human monoclonal antibody from ferrets [56]. There is also a question whether human IgG or a humanized antibody product like SAB185 would bind to ferret FcγR. FcγR binding of IgG has been shown to be important in ADE of dengue, triggering severe dengue viral hemorrhagic fever. A study published in 2021, however, found that ferret FcγR could bind human IgG1 mAb and mediate antibody-dependent cellular cytotoxicity [57]. While we did not evaluate whether ferret FcγR can bind SAB185, this prior finding would lend confidence that at least in ferrets, if low concentrations of SAB185 could trigger ADE we would have seen it. It remains a possibility that SAB185 could trigger ADE in humans or other models, or with other antibody-based treatments of COVID-19.

It was reported that human-derived anti-SARS-CoV-2 antibodies could enhance infection of cells *in vitro* but when these antibodies were infused into mice and macaques they suppressed SARS-CoV-2 replication and did not enhance disease [58]. It is possible that human antibodies do not bind animal FcγRs sufficiently to cause enhanced disease. However, there have been other animal as well as human studies evaluating antibody-based therapies and vaccines and no reports of enhanced disease [58–62]. Based on that and our findings reported here, in addition to the considerable number of human patients that have received monoclonal antibodies or convalescent serum, it seems unlikely that antibody-enhanced disease is a concern for SARS-CoV-2 but further study may be necessary.

## Supporting information

**S1 Fig. No significant elevation in respiratory rate of ferrets infected with WA1.** Ferrets were infected by mucosal or aerosol exposure to WA1. At set intervals after infection, respiratory function was recorded using whole-body plethysmography. Left graph shows percent change in respiratory rate for control, uninfected ferrets compared to mucosally-infected

ferrets (middle graph) and aerosol-infected ferrets (right graph). Graphs show percent change in respiratory rate with baseline (day 0) for individual ferrets in each group. The numbers in the symbol legends are the individual ferret identification numbers.
(TIF)

**S2 Fig. Clinical scores of ferrets infected with WA1 were not significantly elevated relative to controls.** Ferrets were infected by mucosal or aerosol exposure to WA1. At set intervals after infection, with clinical scores recorded twice daily. Left graph shows change in total clinical score for control, uninfected ferrets compared to mucosally-infected ferrets (middle graph) and aerosol-infected ferrets (right graph). The numbers in the symbol legends are the individual ferret identification numbers.
(TIF)

**S3 Fig. No elevation in respiratory rate in ferrets inoculated with SAB-185 and infected with Munich.** Ferrets were inoculated with SAB185 intravenously 1 day prior to challenge with the Munich virus. Graphs show percent change in respiratory rate with baseline (day 0) for individual ferrets in each group. The numbers in the symbol legends are the individual ferret identification numbers.
(TIF)

**S4 Fig. Clinical scores in ferrets inoculated with SAB-185 and then challenged with Munich.** Ferrets were inoculated with SAB185 intravenously 1 day prior to challenge with the Munich virus. Graphs show total clinical scores for individual ferrets in each group. The numbers in the symbol legends are the individual ferret identification numbers.
(TIF)

**S1 Table. Clinical observation scoring.**
(DOCX)

**S2 Table. Plaque assay results for infectious virus in mucosal samples recovered from ferrets after infection with WA1/2020.** Plaque assays were not done on swabs from day 10. Samples are shown as + (positive) or–(negative) rather than a specific titer. Swabs were processed to look for the presence of virus by plaque assay and PCR. Out of concern that virus shedding would be low, near the limit of detection (5 pfu/ml) for plaque assays, plaque assay results from the swabs were scored as positive (+) or negative (-).
(DOCX)

**S3 Table. Fever data for SAB-185-inoculated and control groups.**
(DOCX)

**S4 Table. Plaque assay results for infectious virus in mucosal samples from SAB-185 inoculated and control ferrets after challenge with Munich.** Samples are shown as + (positive) or–(negative) rather than a specific titer. Swabs were processed to look for the presence of virus by plaque assay and PCR. Out of concern that virus shedding would be low, near the limit of detection (5 pfu/ml) for plaque assays, plaque assay results from the swabs were scored as positive (+) or negative (-). ?+ = questionable positive result.
(DOCX)

## Acknowledgments

The views and conclusions contained herein are those of the authors and should not be interpreted as necessarily representing the official policies or endorsements, either expressed or implied, of the U.S. Government.

We would like to thank the staff of the Division of Laboratory Animal Research, the Department of Environmental Health & Safety, and the Regional Biocontainment Laboratory at the University of Pittsburgh for their assistance in conducting these studies. Research was conducted in compliance with the Animal Welfare Act and other Federal statutes and regulations relating to animals and experiments involving animals and adheres to principles stated in the Guide for the Care and Use of Laboratory Animals, National Research Council, 1996. The facility where this research was conducted is fully accredited by the Association for Assessment and Accreditation of Laboratory Animal Care International.

## Author Contributions

**Conceptualization:** Douglas S. Reed, Jeneveve Lundy, Amy L. Hartman, Thomas C. Luke, Kristi Egland, Christoph Bausch, Hua Wu, Eddie J. Sullivan, William B. Klimstra, W. Paul Duprex.

**Data curation:** Douglas S. Reed, Anita K. McElroy, Dominique J. Barbeau, Cynthia M. McMillen, Natasha L. Tilston-Lunel, Shamkumar Nambulli, Emily Cottle, Theron C. Gilliland, Hasala Rannulu, Emily L. Olsen, Katherine J. O'Malley, Amy L. Hartman, Thomas C. Luke, William B. Klimstra.

**Formal analysis:** Douglas S. Reed, Cynthia M. McMillen, Amy L. Hartman, Thomas C. Luke, Christoph Bausch, Hua Wu, William B. Klimstra, W. Paul Duprex.

**Funding acquisition:** Thomas C. Luke, Kristi Egland, Christoph Bausch, Eddie J. Sullivan, William B. Klimstra, W. Paul Duprex.

**Investigation:** Douglas S. Reed, Anita K. McElroy, Dominique J. Barbeau, Cynthia M. McMillen, Natasha L. Tilston-Lunel, Shamkumar Nambulli, Emily Cottle, Theron C. Gilliland, Hasala Rannulu, Jeneveve Lundy, Emily L. Olsen, Katherine J. O'Malley, Mengying Xia.

**Methodology:** Dominique J. Barbeau, Cynthia M. McMillen, Natasha L. Tilston-Lunel, Shamkumar Nambulli, Theron C. Gilliland, Jeneveve Lundy, Emily L. Olsen, Katherine J. O'Malley, Mengying Xia.

**Project administration:** Douglas S. Reed, Thomas C. Luke, Kristi Egland, Christoph Bausch, William B. Klimstra, W. Paul Duprex.

**Supervision:** Douglas S. Reed, Anita K. McElroy, Amy L. Hartman, Thomas C. Luke, William B. Klimstra.

**Writing – original draft:** Douglas S. Reed, Amy L. Hartman.

**Writing – review & editing:** Douglas S. Reed, Anita K. McElroy, Thomas C. Luke, Kristi Egland, Hua Wu, Eddie J. Sullivan, William B. Klimstra, W. Paul Duprex.

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
