## [Decision Letter · Decision Letter 0]

18 Oct 2023

PONE-D-23-26400No evidence for enhanced disease with polyclonal SARS-CoV-2 antibody in the ferret model.PLOS ONE

Dear Dr. Reed,

Thank you for submitting your manuscript to PLOS ONE. After careful consideration, we feel that it has merit but does not fully meet PLOS ONE’s publication criteria as it currently stands. Therefore, we invite you to submit a revised version of the manuscript that addresses the points raised during the review process.

Please submit your revised manuscript by Dec 02 2023 11:59PM. If you will need more time than this to complete your revisions, please reply to this message or contact the journal office at plosone@plos.org. Please include the following items when submitting your revised manuscript:A rebuttal letter that responds to each point raised by the academic editor and reviewer(s). You should upload this letter as a separate file labeled 'Response to Reviewers'.A marked-up copy of your manuscript that highlights changes made to the original version. You should upload this as a separate file labeled 'Revised Manuscript with Track Changes'.An unmarked version of your revised paper without tracked changes. You should upload this as a separate file labeled 'Manuscript'.

We look forward to receiving your revised manuscript.

Kind regards,

Faten Abdelaal Okda

Academic Editor

PLOS ONE

Journal Requirements:

https://journals.plos.org/plospathogens/article?id=10.1371%2Fjournal.ppat.1009308

In your revision ensure you cite all your sources (including your own works), and quote or rephrase any duplicated text outside the methods section. Further consideration is dependent on these concerns being addressed.

"I have read the journal's policy and the authors of this manuscript have the following competing interests: Thomas Luke, Kristi Egland, Christoph Bausch, Hua Wu, Eddie Sullivan are all employed by SAB Biotherapeutics."

7. We notice that your supplementary [Supplemental Tables 1-3] are included in the manuscript file. Please remove them and upload them with the file type 'Supporting Information'. Please ensure that each Supporting Information file has a legend listed in the manuscript after the references list.

Reviewers' comments:

Reviewer's Responses to Questions

**Comments to the Author**

1. Is the manuscript technically sound, and do the data support the conclusions?

Reviewer #1: Partly

Reviewer #2: Partly

Reviewer #3: Yes

2. Has the statistical analysis been performed appropriately and rigorously? 

Reviewer #1: N/A

Reviewer #2: Yes

Reviewer #3: Yes

3. Have the authors made all data underlying the findings in their manuscript fully available?

Reviewer #1: No

Reviewer #2: Yes

Reviewer #3: Yes

4. Is the manuscript presented in an intelligible fashion and written in standard English?

Reviewer #1: Yes

Reviewer #2: Yes

Reviewer #3: Yes

5. Review Comments to the Author

Reviewer #1: The manuscript by Reed et al. submitted to Plos One entitled “No evidence for enhanced disease with polyclonal SARS-CoV-2 antibody in the ferret model.” investigates the potential for antibody-enhanced disease (AED) to SARS-CoV-2 by administering human IgG antibodies into ferrets, followed by virus challenge. The study raises important questions. However, the reviewer feels that the background information/rationale, methods, data presentation, and discussion should be further edited to improve the readability of the manuscript and to support the conclusions drawn.

1. What is known about human IgG (here humanized SAB185 is used) affinity to the ferret FcγR (receptor involved in antibody-dependent cellular cytotoxicity)? While human IgG has great affinity for human FcγR, there are notable species-specific differences in FcγRs. If human IgG has poor affinity for ferret FcγR, then the ferret model may not be a good one to evaluate antibody-enhanced disease in humans. Please provide additional background information for clarity and to support the experimental design and conclusions made here.

a. I suggest editing the title to: No evidence for enhanced disease with human polyclonal SARS-CoV-2 antibody in the ferret model.

2. Phrase “There is also a question of whether SARS-CoV-2 transmission is by contact, droplet, or aerosol” Since aerodynamic size of a respiratory droplet is dynamic, the reviewer suggests avoiding the distinctions of droplet versus aerosol and instead using the term “airborne transmission” (which is inclusive of any size particles traveling through the air) to distinguish it from contact transmission (direct or indirect). PMID: 33977157, 34956601

3. Methods.

a. What were the ages of the ferrets used in this study? Which sexes were used in which experiments?

b. What was the inoculum dose (pfu/ml)?

c. What was the virus diluent used in the nebulizer? What was the humidity and temperature during aerosolization?

d. Please provide the formula used to calculate the inhaled dose.

e. What was used to anesthetize the animals during telemetry surgery? At what site in the abdomen was the implant placed? How much time did the animals have to recover from surgery before infection? Please clarify the details and timeline.

f. Where is the scoring scale included? It’s not in Supplemental Table 1.

g. What was the limit of detection for genomic RNA and PFU? Please specify in methods or figure legends.

4. Results.

a. Why are the actual infectious virus titers not reported in Table S1?

b. Was the fever calculated based on a baseline temperature?

c. Why was viral RNA tissue distribution done for WA1 but not for Munich SCOV2?

d. The figures are at very low resolution, and some text is hard to read.

e. Figs. 2 and 6 should have log10 indicated in the Y axis titles.

f. Fig 3. Is the axis copies/100mg or copies/gram?

g. Please expand and provide more information in the figure legends. The reviewers feels that more information in the legends would allow the reader to understand the graphs without needing to refer to the main text. e.g. Which virus strain and what inoculation route were used in the infection in Figures 4, 5, 6, and 7, this cannot be determined from the legend. What are the numbers associated with symbol legends in figures 4 and 5?

h. Were the WA1/2020 samples collected on days 2, 4, 7, 14 as per table S1? Why is there a day 10 data point in Nasal graph figure 2? And why is not all the data shown? Even if the sample is negative, it could be graphed at or below the limit of detection to show the time course and make it easy to see when the virus was cleared.

i. What groups do the different colors in Fig. 7 represent?

j. What does ?+ mean in table S3?

5. Discussion:

a. “reported here our efforts to develop the ferret as an animal model for COVID-19” The ferret model has been previously used and characterized; other studies report on aerosol inoculations, pathogenesis, transmission data for various SCOV-2 strains, including WA1. The reviewer suggests citing additional relevant manuscripts and discussing the data in the context of similar studies that were conducted in ferrets. For example: PMID: 33827954, 36448801

b. The reviewer appreciates the discussion mentioning influenza virus and studies done in monkeys, but it would be beneficial to contextualize the findings with other studies that conducted aerosol inoculations with both SCOV-2 and flu using the ferret model.

Reviewer #2: Summary: The manuscript describes a study designed to establish and characterize a ferret model of SARS-CoV-2 infection early in the pandemic, comparing two routes of infection (mucosal and aerosol) and ancestral SARS-CoV-2 that emerged in the human population along with an early variant with a mutation in the spike gene. The authors sought to evaluate whether administration of low doses of a polyclonal human IgG antibody (raised in transchromosomic cattle) against the viral spike protein would cause enhanced disease upon subsequent viral challenge.

Comments: The study seeks to provide important and relevant information (particularly given it was performed early in the pandemic) understanding characteristics of the ferret model and the potential for AED in SARS-CoV-2. The authors address some limitations such as the discrepancy in dose, but for publication the manuscript needs extensive revision to address the following:

• The introduction was nicely written and describes rationale for the study.

• Methods section – would be ideal to provide some more key details even if referred to previously published methods. For example, what was the target for qRT-PCR (so reader doesn’t need to chase down previous manuscripts) and what controls were employed – useful to discuss what negative controls were performed (such as from uninfected ferret tissues) to provide confidence in validity of PCR data and whether ‘genomes/ml or /gram’ is appropriate metric based on what being measured.

• Need to define what body temperature is defined as a ‘fever’ in ferrets or just describe as spike in temperature (not spike in fever)

• Although the SAB-185 antibody was previously described/characterized, some key details need to be described here as they relate to the study. Here we see no data or description of binding characteristics of this antibody to spike and if neutralizing and the potency that would allow the reader to evaluate whether it is an appropriate candidate for examining AED. It is mentioned offhand once at end of the introduction that this antibody is neutralizing. There is no data shown or described regarding the relationship between levels of antibody administered and virus neutralization in serum (or to show the doses given are sub-neutralizing). There is no description of how the choice of a human IgG might promote AED versus a ferret IgG given some of the known mechanisms of AED.

• Clinical score and plethysmography were both mentioned in methods but findings were not described nor any data shown.

• Given a focus of the manuscript is enhancement of disease/pathology seen in other examples of respiratory viruses, the pathologic characterization or description of it was inadequate. Need to better discuss what the pathologic changes observed were, if there were inflammatory infiltrates and if these differed in any way in the SAB-185 treated animals.

• Given pathology is not uniform in lungs, need to provide better description of how this was accounted for in regions selected for alveolar space analysis. Similarly, what did the histopathology images shown represent or why were these selected – were these representative of the most severe pathology in each group?

Reviewer #3: Reed et al have prepared a well written and clearly presented account of SARS-CoV-2 animal model development, and application to evaluation of an immunotherapy.

The number of animals used is small, which needs to be taken into account in the interpretation of the results, but these data are valuable and informative to the field.

The reprographic quality of the figures is poor, particularly affecting Figure 6 and negating the ability to discern pathology in Figure 7.

Minor typographical issues:

Capital M2 on page 19 to replace ‘M1 and m2’

Figure 3 – ‘aerosol’ label is split over 2 rows

Page 20: ‘but with a European isolate that had the D614G isolate’. Second ‘isolate’ should be ‘mutation’?

Minor issues with interpretation of the results:

Page 18: ‘The two mucosally-inoculated ferrets (designated M1 & M2) received a dose of 4x106 pfu; both had a spike in fever within the first 36 hours of infection...’. It is not clear from the figure where M2 has a temperature spike. Consider improving the clarity/visibility of the pink line. Also, provide an explanation for the arrow.

Page 19: ‘Infectious virus was found in the oral and nasal cavities from all four aerosol-infected ferrets on day 2’. Not all samples were positive, suggest changing to ‘oral and/or nasal’.

Page 20: ‘The following morning, all of the ferrets were challenged with 8x105 pfu of Munich-1.1 by mucosal inoculation (IN/oral).’ Please clarify if infection was not as described in the methods which state mucosal inoculation is intranasal, oral and in the eye.

Page 21. ‘Infectious virus was also found in the oral swabs and nasal washes from ferrets in all three groups out to day 8 post-infection (Supplemental Table 3)’. From the table it appears it is all 3 groups out to day 6 and a single positive in the control group at day 8.

Most significant comment:

Page 24: ‘Our results (fever, weight, viral shedding, pathology), however, would suggest that at least with SAB185 in ferrets at the doses administered, there are no concerns about ADE’.

Regarding 'there are no concerns'. Given the low number of animals used, and the considerable variation in arguably the most important readout (lung pathology/alveolar space) the study is only powered to detect a very large enhancement of disease. The wording in the results ‘no evidence for antibody-enhancement of lung pathology’ is more appropriate. I would propose the discussion applies similar.

6. PLOS authors have the option to publish the peer review history of their article (what does this mean?). If published, this will include your full peer review and any attached files.

Reviewer #1: No

Reviewer #2: No

Reviewer #3: No

---

## [Author Response · Author response to Decision Letter 0]

14 Mar 2024

https://journals.plos.org/plospathogens/article?id=10.1371%2Fjournal.ppat.1009308

In your revision ensure you cite all your sources (including your own works), and quote or rephrase any duplicated text outside the methods section. Further consideration is dependent on these concerns being addressed.

Can you be more specific about where the overlapping text is? Other than the ethics statement at the start of the Materials & Methods section – which is boilerplate language - we could not find any overlapping test using the admittedly limited software that we have to compare the two. We do take this seriously and are happy to make changes where warranted, we are just not able to see what is similar between the two outside of the methods section. It should be noted that the paper mentioned includes myself and other members of my laboratory as co-authors. Considering we have not previously published on ferrets, and the other paper is about a different pathogen in cynomolgus macaques using EEG & ICP telemetry (which is not used in this study), it is hard to conceive of any overlap with this other manuscript outside of the methods section.

Done. As stated in the Acknowledgements section of the paper, the funding for these studies was provided by the Department of Defense (DOD) and the Biomedical Advanced Research Development Authority (BARDA), not SAB Biotherapeutics. We have made sure our statements in the Funding Information and Financial Disclosure sections reflect this accurately. We did note that in the Declaration of Interests it stated that the work was supported by a DOD contract with SAB. The University of Pittsburgh was contracted by SAB for the animal/virus work, but the money came from the DOD. This has been updated so as to more accurately reflect the funding source.

"I have read the journal's policy and the authors of this manuscript have the following competing interests: Thomas Luke, Kristi Egland, Christoph Bausch, Hua Wu, Eddie Sullivan are all employed by SAB Biotherapeutics."

Done

Data will be uploaded to the Open Science Framework repository (OSF), DOI number >>>>>>

There were no human subjects in this study. As stated in our ethics section, this work was approved by the University of Pittsburgh’s Institutional Animal Care and Use Committee (IACUC). The polyclonal human antibody product used was made in transgenic cows with human immunoglobulin genes, not obtained from humans.

7. We notice that your supplementary [Supplemental Tables 1-3] are included in the manuscript file. Please remove them and upload them with the file type 'Supporting Information'. Please ensure that each Supporting Information file has a legend listed in the manuscript after the references list.

Done

Reviewers' comments:

Reviewer's Responses to Questions

Comments to the Author

1. Is the manuscript technically sound, and do the data support the conclusions?

Reviewer #1: Partly

Reviewer #2: Partly

Reviewer #3: Yes

2. Has the statistical analysis been performed appropriately and rigorously? 

Reviewer #1: N/A

Reviewer #2: Yes

Reviewer #3: Yes

3. Have the authors made all data underlying the findings in their manuscript fully available?

Reviewer #1: No

Reviewer #2: Yes

Reviewer #3: Yes

4. Is the manuscript presented in an intelligible fashion and written in standard English?

Reviewer #1: Yes

Reviewer #2: Yes

Reviewer #3: Yes

5. Review Comments to the Author

Reviewer #1: The manuscript by Reed et al. submitted to Plos One entitled “No evidence for enhanced disease with polyclonal SARS-CoV-2 antibody in the ferret model.” investigates the potential for antibody-enhanced disease (AED) to SARS-CoV-2 by administering human IgG antibodies into ferrets, followed by virus challenge. The study raises important questions. However, the reviewer feels that the background information/rationale, methods, data presentation, and discussion should be further edited to improve the readability of the manuscript and to support the conclusions drawn.

1. What is known about human IgG (here humanized SAB185 is used) affinity to the ferret FcγR (receptor involved in antibody-dependent cellular cytotoxicity)? While human IgG has great affinity for human FcγR, there are notable species-specific differences in FcγRs. If human IgG has poor affinity for ferret FcγR, then the ferret model may not be a good one to evaluate antibody-enhanced disease in humans. Please provide additional background information for clarity and to support the experimental design and conclusions made here.

We found one paper that examined ferret FcγR binding a human IgG1 mAb, evaluating ADCC enhancement across several species. We also found a paper describing clearance of human influenza mAb within a week of inoculation. However, SAB185 is a polyclonal antibody product so all four IgG subtypes could be represented. We’ve added all of this to the discussion as well as what is known about ADE in dengue fever. We have added to the discussion to say that while in the ferrets SAB185 did not trigger ADE, it remains possible that it would be seen in other models or in humans, or with other antibody treatments for COVID-19. 

a. I suggest editing the title to: No evidence for enhanced disease with human polyclonal SARS-CoV-2 antibody in the ferret model.

Done. We agree and thank the reviewer for the suggestion.

2. Phrase “There is also a question of whether SARS-CoV-2 transmission is by contact, droplet, or aerosol” Since aerodynamic size of a respiratory droplet is dynamic, the reviewer suggests avoiding the distinctions of droplet versus aerosol and instead using the term “airborne transmission” (which is inclusive of any size particles traveling through the air) to distinguish it from contact transmission (direct or indirect). PMID: 33977157, 34956601

We have re-written the sentence and replaced droplet/aerosol with airborne, as the reviewer suggested.

3. Methods.

a. What were the ages of the ferrets used in this study? Which sexes were used in which experiments?

Ferrets were young adults (4-7 months in age). In the initial experiment, all ferrets were male. In the second experiment, they were half female, half male. We have added a column to Supplemental Table 2 and added the sex of the ferrets.

b. What was the inoculum dose (pfu/ml)? It was 4x106 pfu for WA1, 8x105 for Munich. We have added this to the mucosal inoculation section of the methods. For WA1 by aerosol, it was 2.3x104 pfu, which was added to the aerosol exposure section.

c. What was the virus diluent used in the nebulizer? What was the humidity and temperature during aerosolization? Virus was in Opti-MEM, although it was not diluted for aerosolization. AGI samples were collected in Opti-MEM with Antifoam A. We have added Opti-MEM in the virology and aerosol exposure sections.

d. Please provide the formula used to calculate the inhaled dose. As stated in the aerosol exposure section of the methods “…inhaled dose was calculated as the product of aerosol concentration of the virus and the accumulated volume of inhaled air, determined as the product of the duration of exposure and the animal’s minute volume.”

e. What was used to anesthetize the animals during telemetry surgery? At what site in the abdomen was the implant placed? How much time did the animals have to recover from surgery before infection? Please clarify the details and timeline. Ferrets were allowed to heal for 2-4 weeks after surgery. We have added this to the methods section.

f. Where is the scoring scale included? It’s not in Supplemental Table 1. 

Added and renumbered Supplemental Tables accordingly.

g. What was the limit of detection for genomic RNA and PFU? Please specify in methods or figure legends. The limit of detection for genomic RNA was 1,856 genomes/ml, which we have added to the figure legends as requested. Similarly, the limit of detection for pfu was 5 pfu/ml, which we have added to the legend for Supplemental Tables 2 and 4.

4. Results.

a. Why are the actual infectious virus titers not reported in Table S1? Swabs were processed to look for the presence of virus by plaque assay and PCR. Because we thought that virus shedding would be low, at the limit of detection for plaque assays, we scored the plaque assay results from the swabs as positive (+) or negative (-). We have added this statement to Supplemental Table 2 (Formerly Supplemental Table 1).

b. Was the fever calculated based on a baseline temperature?

Yes, as detailed in the methods section. We have added additional details on how fever was defined.

c. Why was viral RNA tissue distribution done for WA1 but not for Munich SCOV2?

Only a few tissues from a couple of ferrets were positive for virus after infection with Munich.

d. The figures are at very low resolution, and some text is hard to read. Figures provided were TIFF files at 600 dpi. Not sure why they appeared to be low resolution or hard to read. We do agree that in the assembled pdf they are not very nice to look at.

e. Figs. 2 and 6 should have log10 indicated in the Y axis titles.

Done.

f. Fig 3. Is the axis copies/100mg or copies/gram?

We have corrected the legend to reflect what was shown in the figure, which was accurate – copies/100 mg.

g. Please expand and provide more information in the figure legends. The reviewers feels that more information in the legends would allow the reader to understand the graphs without needing to refer to the main text. e.g. Which virus strain and what inoculation route were used in the infection in Figures 4, 5, 6, and 7, this cannot be determined from the legend. What are the numbers associated with symbol legends in figures 4 and 5?

We have made the requested changes, including adding the virus strain and route to Figures 4-7 and indicating that the symbol legends in figures 4 and 5 denote the ferret identification number.

h. Were the WA1/2020 samples collected on days 2, 4, 7, 14 as per table S1? Why is there a day 10 data point in Nasal graph figure 2? And why is not all the data shown? Even if the sample is negative, it could be graphed at or below the limit of detection to show the time course and make it easy to see when the virus was cleared.

Swabs were collected on days 2, 4, 7, 10, and 14. We have added this information to the legends of the figures so it is clear. Samples not plotted had zero plaques or genomes, which cannot be plotted on a log scale graph.

i. What groups do the different colors in Fig. 7 represent?

The different colors represent individual ferrets within the group; we have added that to the legend.

j. What does ?+ mean in table S3? That means it was a questionable positive result. We have added that to the legend for the table (which is now S4).

5. Discussion:

a. “reported here our efforts to develop the ferret as an animal model for COVID-19” The ferret model has been previously used and characterized; other studies report on aerosol inoculations, pathogenesis, transmission data for various SCOV-2 strains, including WA1. The reviewer suggests citing additional relevant manuscripts and discussing the data in the context of similar studies that were conducted in ferrets. For example: PMID: 33827954, 36448801

b. The reviewer appreciates the discussion mentioning influenza virus and studies done in monkeys, but it would be beneficial to contextualize the findings with other studies that conducted aerosol inoculations with both SCOV-2 and flu using the ferret model.

As regards both points, we have re-written the first paragraph in the discussion to focus more on the ferret model of COVID-19 and what others have reported in the published literature.

Reviewer #2: Summary: The manuscript describes a study designed to establish and characterize a ferret model of SARS-CoV-2 infection early in the pandemic, comparing two routes of infection (mucosal and aerosol) and ancestral SARS-CoV-2 that emerged in the human population along with an early variant with a mutation in the spike gene. The authors sought to evaluate whether administration of low doses of a polyclonal human IgG antibody (raised in transchromosomic cattle) against the viral spike prote

---

## [Editor Report · Decision Letter 1]

7 May 2024

No evidence for enhanced disease with human polyclonal SARS-CoV-2 antibody in the ferret model.

PONE-D-23-26400R1

Dear Dr. Reed,

We’re pleased to inform you that your manuscript has been judged scientifically suitable for publication and will be formally accepted for publication once it meets all outstanding technical requirements.

Kind regards,

Faten Abdelaal Okda

Academic Editor

PLOS ONE
---

## [Editor Report · Acceptance letter]

27 May 2024

PONE-D-23-26400R1 

PLOS ONE

Dear Dr. Reed, 

I'm pleased to inform you that your manuscript has been deemed suitable for publication in PLOS ONE. Congratulations! Your manuscript is now being handed over to our production team.

Kind regards, 

on behalf of

Dr. Faten Abdelaal Okda 

Academic Editor

PLOS ONE